# Elevated Plasma Soluble C-Type Lectin-like Receptor 2 Is Associated with the Worsening of Coronavirus Disease 2019

**DOI:** 10.3390/jcm11040985

**Published:** 2022-02-14

**Authors:** Hideo Wada, Yuhuko Ichikawa, Minoru Ezaki, Akitaka Yamamoto, Masaki Tomida, Masamichi Yoshida, Shunsuke Fukui, Isao Moritani, Katsuya Shiraki, Motomu Shimaoka, Toshiaki Iba, Katsue Suzuki-Inoue, Hideto Shimpo

**Affiliations:** 1Department of General and Laboratory Medicine, Mie Prefectural General Medical Center, Yokkaichi 510-0885, Japan; katsuya-shiraki@mie-gmc.jp; 2Department of Central Laboratory, Mie Prefectural General Medical Center, Yokkaichi 510-0885, Japan; ichi911239@yahoo.co.jp (Y.I.); ajbyd06188@yahoo.co.jp (M.E.); 3Department of Emergency and Critical Care Center, Mie Prefectural General Medical Center, Yokkaichi 510-0885, Japan; akitaka-yamamoto@mie-gmc.jp (A.Y.); st25053@yahoo.co.jp (M.T.); 4Department of Respiratory Medicine, Mie Prefectural General Medical Center, Yokkaichi 510-0885, Japan; masamichi-yoshida@mie-gmc.jp; 5Department of Gastroenterology, Mie Prefectural General Medical Center, Yokkaichi 510-0885, Japan; m13092sf@jichi.ac.jp (S.F.); isao-moritani@mie-gmc.jp (I.M.); 6Department of Molecular Pathobiology and Cell Adhesion Biology, Mie University Graduate School of Medicine, Tsu 514-8507, Japan; motomushimaoka@gmail.com; 7Department of Emergency and Disaster Medicine, Juntendo University Graduate School of Medicine, Tokyo 113-8421, Japan; toshiiba@juntendo.ac.jp; 8Department of Clinical and Laboratory Medicine, Faculty of Medicine, University of Yamanashi, Yamanashi 409-3898, Japan; katsuei@yamanashi.ac.jp; 9Mie Prefectural General Medical Center, Yokkaichi 510-0885, Japan; hideto-shimpo@mie-gmc.jp

**Keywords:** COVID-19, coagulopathy, platelet activation, sCLEC-2

## Abstract

Although thrombosis in coronavirus disease 2019 (COVID-19) infection has attracted attention, the mechanism underlying its development remains unclear. The relationship between platelet activation and the severity of COVID-19 infection was compared with that involving other infections. Plasma soluble C-type lectin-like receptor 2 (sCLEC-2) levels were measured in 46 patients with COVID-19 infection and in 127 patients with other infections. The plasma sCLEC-2 levels in patients with COVID-19 infection {median (25th, 75th percentile), 489 (355, 668) ng/L} were significantly higher (*p* < 0.001) in comparison to patients suffering from other pneumonia {276 (183, 459) ng/L}, and the plasma sCLEC-2 levels of COVID-19 patients with severe {641 (406, 781) ng/L} or critical illness {776 (627, 860) ng/L} were significantly higher (*p* < 0.01, respectively) in comparison to those with mild illness {375 (278, 484) ng/L}. The ratio of the sCLEC-2 levels to platelets in COVID-19 patients with critical illness of infection was significantly higher (*p* < 0.01, *p* < 0.001 and *p* < 0.05, respectively) in comparison to COVID-19 patients with mild, moderate or severe illness. Plasma sCLEC-2 levels were significantly higher in patients with COVID-19 infection than in those with other infections, suggesting that platelet activation is triggered and facilitated by COVID-19 infection.

## 1. Introduction

Since its initial outbreak in China [1,2], coronavirus disease 2019 (COVID-19) has spread worldwide, affecting Europe, North and South America, Asia and Africa, resulting in a pandemic [3]. Approximately 2% of patients with COVID-19 die, and 5–10% of patients develop severe and life-threatening acute respiratory distress syndrome (ARDS) [4,5,6], with many more patients suffering or moderate infectious disease [7]. The relationship between COVID-19 infection and thrombosis has attracted attention [8]. Various biomarkers such as D-dimer have been reported as specific biomarkers of the severity of COVID-19 infection but are not specific for COVID-19 infection itself [9,10]. 

Several mechanisms underlying the worsening of the condition of COVID-19 patients, such as cytokine storm [11], primary pulmonary thrombosis, vascular endothelial injuries [12], and platelet activation [13], have been proposed [14]. One of the clinical features of COVID-19 is the high prevalence of arterial thrombosis such as stroke and ischemic heart disease, and we speculated that the activation in platelets is involved in the pathogenesis. Soluble C-type lectin-like receptor 2 (sCLEC-2) has been introduced as a new biomarker of platelet activation [15]. Elevated plasma levels of sCLEC-2 have been reported in patients with thrombotic microangiopathy (TMA) and disseminated intravascular coagulation (DIC) [16,17], as well as in patients with acute coronary syndrome [18] or acute cerebral infarction [19].

In this study, the levels of plasma biomarkers, such as sCLEC-2 and D-dimer, in 46 patients with COVID-19 were measured and compared to those in 127 patients with other infections to determine the mechanism underlying the worsening of COVID-19 infection.

## 2. Materials and Methods

Forty-six COVID-19 patients (mild illness [individuals with any of the various signs and symptoms of COVID-19 such as fever, cough, sore throat, headache, strong fatigue, muscle and body aches, loss of taste or smell, etc.], *n* = 16; moderate illness [individuals who show evidence of lower respiratory disease during clinical assessment or imaging and who have an oxygen saturation (SpO_2_) ≥94% on room air at sea level], *n* = 9; severe illness [individuals who have SpO_2_ < 94% on room air at sea level, a ratio of arterial partial pressure of oxygen to fraction of inspired oxygen (PaO_2_/FiO_2_) < 300 mm Hg, a respiratory rate >30 breaths/min, or lung infiltration >50%.], *n* = 16; critical illness [individuals who have respiratory failure, septic shock, and/or multiple organ dysfunction], *n* = 5 [20], were admitted to Mie Prefectural General Medical Center from 1 March to 31 July, 2021. Regarding other infections who were managed at Mie Prefectural General Medical Center from 1 September 2019 to 28 December 2020, 53 patients with other pneumonia, 17 with other upper respiratory tract infection without pneumonia, 18 with sepsis, 14 with biliary tract infections, and 25 with urinary tract infections were also examined (Table 1 and Table 2). Sixty patients with unidentified clinical syndrome (UCS) were examined as controls. DIC was diagnosed using the diagnostic criteria for DIC established by the Japanese Ministry of Health, Labor and Welfare [21]. 

Among patients with COVID-19 infection, steroids or anti-viral agents were used for the treatment of patients with moderate illness, while oxygen therapy was used for patients with severe illness. Patients with critical illness received artificial respirator management in the intensive care unit (ICU). Patients with bacterial infection were treated with antibiotics and the severe cases were managed in the ICU. 

The study protocol (O-0057) was approved by the Human Ethics Review committees of Mie Prefectural General Medical Center, and informed consent was obtained from each patient. 

COVID-19 infection was diagnosed by polymerase chain reaction using the Gene expert system (Cepheid, CA, USA). We measured the plasma sCLEC-2 levels via chemiluminescent enzyme immunoassay (CLEIA) using previously described monoclonal antibodies and the STACIA CLEIA system (LSI Medience, Tokyo, Japan) [17,19]. In brief, magnetic particles were coated with the anti-CLEC-2 monoclonal antibody 11D5. The plasma samples were then incubated with antibody-coated magnetic particles, and after being washed, they were incubated further with the alkaline-phosphatase-labeled anti-CLEC-2 monoclonal antibody 11E6. After washing again, the magnetic particles were incubated with chemiluminescent substrate solution (CDP-Star; Applied BioSystems, Waltham, MA, UAS) and luminescence was measured using the luminometer installed in the STACIA system. The D-dimer levels were determined via the latex agglutination method using LPIA-Genesis (LSI Medience) [22]. The fibrinogen levels and prothrombin time (PT) -international normalize ratio (INR) were measured using an automatic coagulation analyzer (CS-5100; Sysmex, Kobe, Japan) using a Thrombocheck Fib (L) and Thromborel S (Siemens Healthcare Diagnostics Products, Malvern, PA, USA). 

### 2.1. Study Design 

(a)Sex, age, mortality, frequency of association with DIC, laboratory data (e.g., platelet count, PT-INR, and D-dimer levels in patients with COVID-19 infection were compared with those in patients suffering from other infections.(b)Comparison of plasma sCLEC-2 levels between patients with COVID-19 infection and patients suffering with other infections. (c)Sex, age, mortality and laboratory data were examined among the four clinical stages of COVID-19 infection.(d)The plasma sCLEC-2 levels and sCLEC-2/platelets ratio were examined among the four clinical stages of COVID-19 infection.

### 2.2. Statistical Analyses

The data are expressed as the median (25th, 75th percentile). Differences between independent groups were examined using the Mann-Whitney *U-test*. *p*-values of ≤0.05 were considered to indicate statistical significance. All statistical analyses were performed using the Stat flex software program (version 6; Artec Co., Ltd., Osaka, Japan).

## 3. Results

Patients with other infections were older in comparison to patients with COVID-19 infection (Table 1). The mortality rate in patients suffering from other pneumonia, sepsis and COVID-19 infection was 17.0%, 16.7% and 4.3%, respectively. Among the patients with COVID-19 infection, one patient died due to gastrointestinal bleeding and one died due to senility. 

Platelet counts were significantly lower in patients with sepsis, biliary tract infections or urinary tract infections than in those with UCS. The PT-INR was significantly higher in patients with all infections in comparison to patients with UCS. The plasma D-dimer levels were significantly higher in patients with other pneumonia, sepsis, biliary tract or urinary tract infections than in patients with UCS or COVID-19 patients (Figure 1). There were no significant differences in D-dimer levels between patients with UCS and those with COVID-19.

The plasma sCLEC-2 levels were significantly higher in patients with all infections than in patients with UCS, as well as in patients with COVID-19 infection {489 (355, 668) ng/L} than in patients with all other infections (Figure 2).

Regarding patients with COVID-19 infection (Table 2), one patient with mild stage and one with severe illness died, but not due to COVID-19. There were no significant differences in platelet counts (Figure 3a) or the PT-INR among the four stages of COVID-19 infection. Plasma D-dimer levels were significantly higher in patients with severe or critical illness than in those with mild illness of COVID-19 infection. Plasma sCLEC-2 levels were significantly higher in patients with severe {641 (406, 781) ng/L} or critical illness {776 (627, 860) ng/L} than in those with mild illness {375 (278–484) ng/L} (Figure 3b). The sCLEC-2-to-platelets ratio was significantly higher in patients with critical illness {45.6 (43.0, 52.3)} than in those with mild {15.2 (13.4–28.8)}, moderate {27.9 (19.7–28.8)}, or severe illness {45.6 (43.0–52.3)} and higher in patients with severe illness than in those with mild illness of COVID-19 infection (Figure 3c).

## 4. Discussion

There have been many reports concerning the relationship between thrombosis and COVID-19 infection [23,24], and several mechanisms underlying thrombosis in COVID-19 infection such as local pulmonary thrombogenesis [12], hyper-coagulable state (including activation of tissue factor pathway) [25], inflammation induced by inflammatory cytokines [11], platelet activation [13] and neutrophil extracellular traps (NETs) [26], have been proposed.

Although elevated D-dimer levels are a well-known risk factor for thrombosis in COVID-19 infection [27], markedly high D-dimer levels were not observed in patients with COVID-19 infection in the present study. As a matter of fact, there were no complications of thrombosis or DIC in any patients with COVID-19 infection in this study; in contrast, those with other pneumonia or sepsis who had markedly elevated D-dimer levels were often associated with DIC. However, slightly high levels of D-dimer were observed in the patients with severe or critical illness of COVID-19 infection, suggesting that D-dimer elevation is recognized as be the result of COVID-19-associated coagulopathy.

The platelet count is determined by the balance among the platelet consumption, destruction and production rates. As a number of different viruses interfere with hematopoiesis, thrombocytopenia is a common phenomenon seen in various virus infections, including COVID-19 [28]. In the present study, thrombocytopenia was observed in patients with critical illness of COVID-19 infection as well as those with sepsis who were associated with DIC. Since the balance of coagulation disorder and thrombocytopenia has been reported to be disproportionate between COVID-19 and sepsis, the pathogenesis of the coagulopathies should be different. The systemic activation in coagulation and the suppressed fibrinolysis is the major mechanism underlying thrombocytopenia in bacterial sepsis [29]. Meanwhile, such coagulation/fibrinolytic disorder is observed only at the end-stage in COVID-19. On the other hand, arterial thrombosis often complicates during the course of COVID-19 and the involvement of platelet activation should exist from the earlier stage. In COVID-19, platelets are usually activated without thrombocytopenia similar to local thromboses such as acute coronary syndrome [18] or acute cerebral infarction [19]. We speculate the spike protein-induced platelet activation via the angiotensin converting enzyme 2 play a role [30]. In this scenario, thrombosis may attribute platelet factor 4, von Willebrand factor, and other substances released from platelets [31]. 

Soluble forms of CLEC2 are released upon platelet activation and the elevated sCLEC-2 levels were reported in patients with acute coronary syndrome [18] or acute cerebral infarction [19] without thrombocytopenia, suggesting that sCLEC2 levels may reflect platelet activation in atherosclerotic thrombosis. In the present study, the plasma sCLEC-2 levels in patients with COVID-19 infection were significantly higher than in those with other infections and reflected the severity of COVID-19 infection. In particular, the sCLEC-2-to-platelet ratio is useful for evaluating the severity of COVID-19 infection. Recently, the sCLEC-2-to-platelet ratio is reported to be useful for early diagnosis of sepsis-induced coagulopathy [32]. Furthermore, the plasma sCLEC-2 levels in patients with mild illness of COVID-19 infection were similar to those in patients with other pneumonia. These findings suggest that the activation of platelets may occur in early COVID-19 infection without DIC. A low dose of aspirin was reported useful for managing COVID-19 infection [33].

Finally, Plasma sCLEC-2 levels were significantly higher in patients with COVID-19 infection than in those with bacterial infections. On the other hand, plasma D-dimer levels were significantly higher in patients with bacterial infections than COVID-19 infection. These findings suggest that COVID-19 infection tends to facilitate platelet activation, while bacterial infections tend to facilitate fibrin generation (Figure 4).

## 5. Limitations and Further Study

The present study, was relatively small in scale, only included a small number of COVID-19 patients with critical illness and did not match the COVID-19 group and other infections group for age and sex. Further large-scale studies should be conducted with a control group that is matched for age and sex. In addition to biomarkers of platelet activation, biomarkers of vascular endothelial injury should be examined.

## 6. Conclusions

As the plasma sCLEC-2 levels were significantly elevated in patients with COVID-19 infection and reflected the severity of COVID-19 infection, the measurement of sCLEC-2 level may be useful for predicting the outcome of COVID-19 infection.

## Figures and Tables

**Figure 1 jcm-11-00985-f001:**
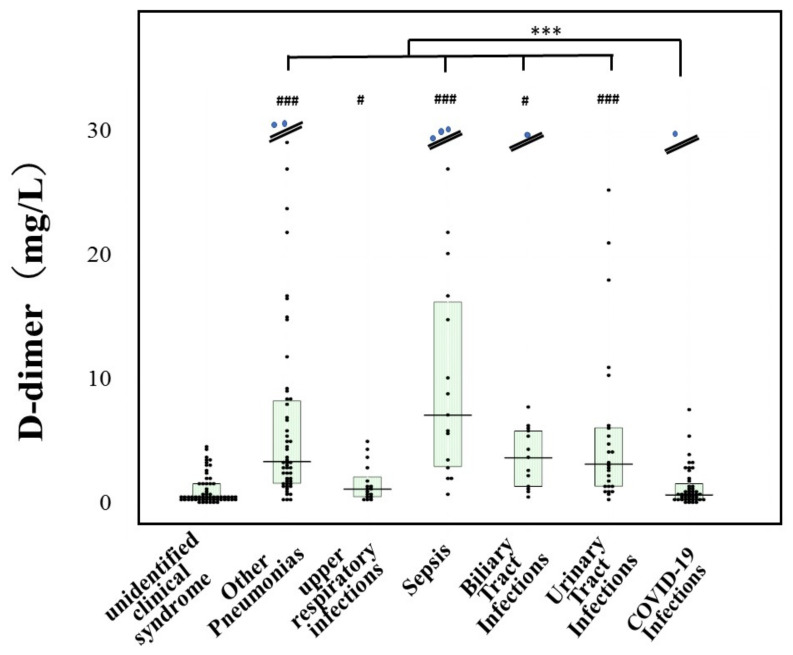
Plasma levels of D-dimer in various infections and COVID-19 infection. ***, *p* < 0.001 in comparison with COVID-19 infection; ^#^, *p* < 0.05 in comparison with unidentified clinical syndrome; ^###^, *p* < 0.001 in comparison with unidentified clinical syndrome.

**Figure 2 jcm-11-00985-f002:**
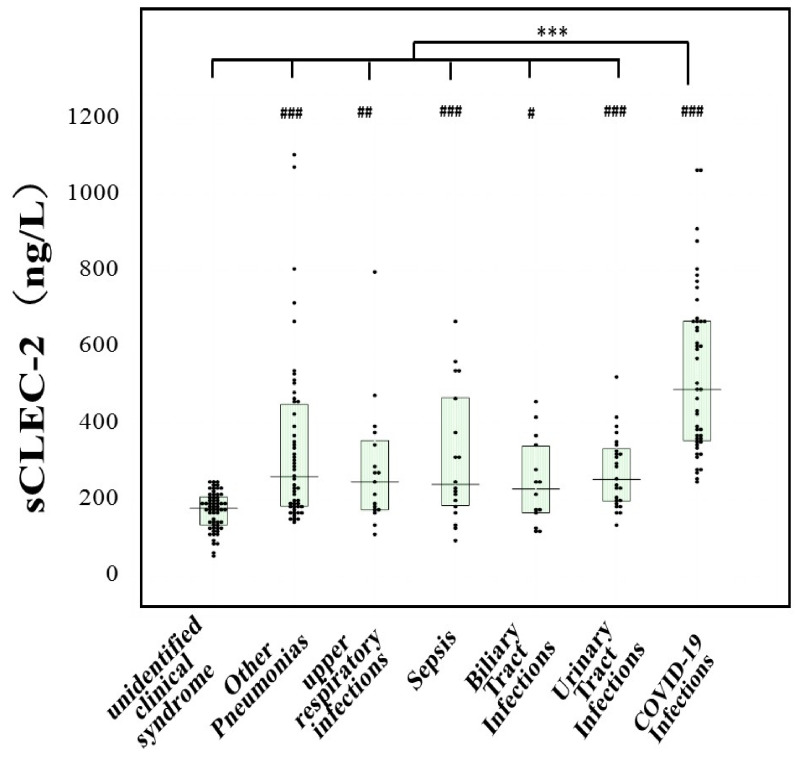
Plasma levels of sCLEC-2 in various infections and COVID-19 infection ***, *p* < 0.001 in comparison with COVID-19 infection; ^#^, *p* < 0.05 in comparison with unidentified clinical syndrome; ^##^, *p* < 0.01 in comparison with unidentified clinical syndrome; ^###^, *p* < 0.001 in comparison with unidentified clinical syndrome.

**Figure 3 jcm-11-00985-f003:**
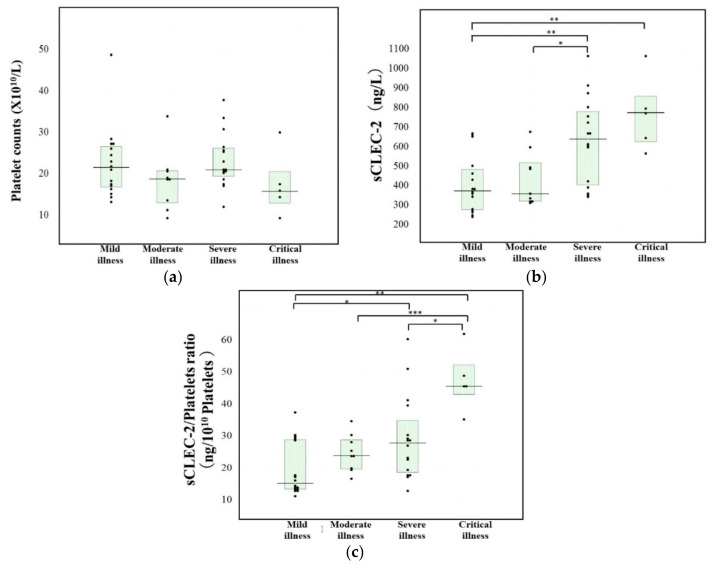
Platelets (**a**), sCLEC-2 (**b**) and the sCLEC-2/Platelets ratio (**c**) in patients with COVID-19 infection ***, *p* < 0.001; **, *p* < 0.01; *, *p* < 0.05.

**Figure 4 jcm-11-00985-f004:**
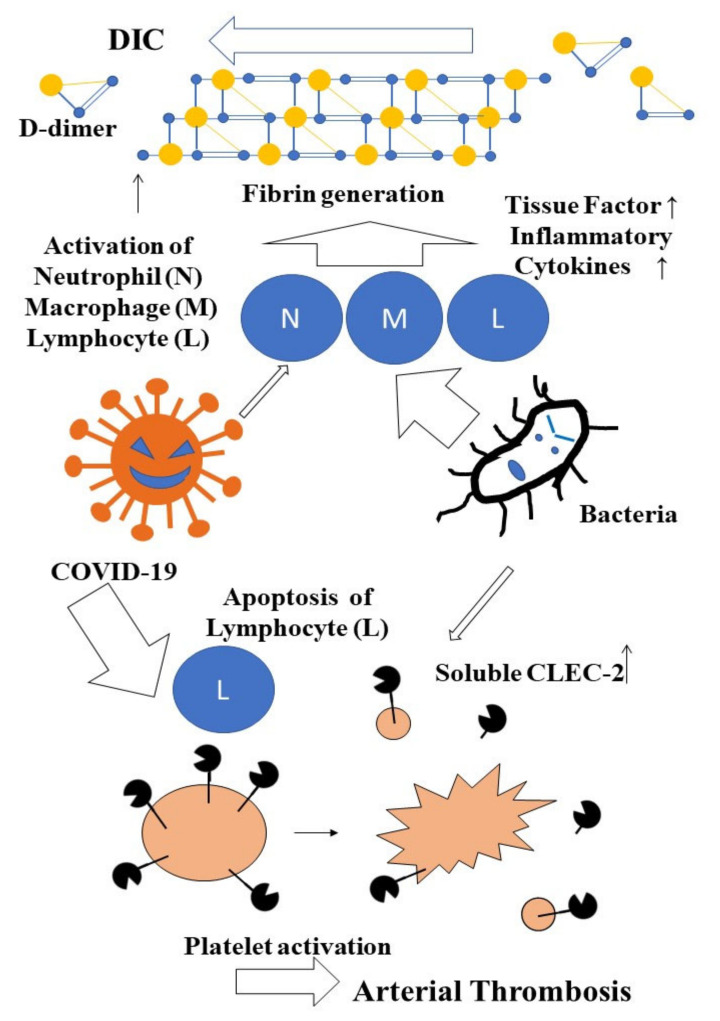
Difference of sCLEC-2 and D-dimer levels between COVID-19 and bacterial infections.

**Table 1 jcm-11-00985-t001:** Patients with various infections.

Underlying Diseases	N	F (%):M	Age (Years)	Death (%)	DIC	Platelets (×10^10^/L)	PT-INR	D-Dimer (mg/L)
Unidentified Clinical Syndrome	60	31 (51.7%):29	56.5 ^#^ (48.0–75.0)	0	0 (0)	23.1 (17.9–27.0)	0.96 ^###^ (0.91–1.00)	0.5 (0.4–1.6)
Other Pneumoniae	53	23 (43.4%):30	81.0 ***^###^ (71.0–85.0)	9 (17.0)	6 (11.3)	25.4 (17.7–27.7)	1.12 ***^##^ (1.05–1.20)	3.4 ***^###^ (1.8–8.7)
Upper Respiratory Infections	17	10 (58.8%):7	71.0 *^#^ (50.3–77.3)	0 (0)	0 (0)	25.4 (17.7–27.7)	1.03 ** (0.96–1.09)	1.2 * (0.6–2.2)
Sepsis	18	7 (38.9%):11	67.0 ***^#^ (49.0–83.0)	3 (16.7)	7 (38.9)	13.7 **^#^ (8.3–21.3)	1.20 ***^###^ (1.11–1.42)	9.5 ***^###^ (3.6–21.9)
Biliary Tract Infections	14	8 (57.1%):6	84.5 ***^###^ (76.0–92.0)	0 (0)	1 (14.3)	16.6 * (8.5–26.6)	1.05 *** (0.96–1.15)	4.1 ***^###^ (1.5–6.1)
Urinary Tract Infections	25	18 (72.0%):7	79.0 ***^###^ (67.8–86.0)	0 (0)	0 (4.0)	20.7 (17.2–25.8) *	1.08 *** (1.01–1.13)	3.2 ***^###^ (1.4–6.1)
COVID-19 Infections	46	21 (45.7%):25	51.0 (28.0–67.0)	2 (4.3)	0 (0)	20.7 (17.2–25.8)	1.04 *** (1.01–1.08)	0.8 (0.4–1.8)

Data was shown as median (25–75 percentile). DIC, disseminated intravascular coagulation; COVID, coronavirus infectious disease; PT-INR, prothrombin time–international normalized ratio; ***, *p* < 0.001 in comparison with unidentified clinical syndrome; **, *p* < 0.01 in comparison with unidentified clinical syndrome; *, *p* < 0.05 in comparison with unidentified clinical syndrome; ^###^, *p* < 0.001 in comparison with COVID-19 infections; ^##^, *p* < 0.01 in comparison with COVID-19 infections; ^#^, *p* < 0.05 in comparison with COVID-19 infections.

**Table 2 jcm-11-00985-t002:** Patients with COVID-19 infections.

	N	F:M	Age (Years)	Death (%)	Platelets (×10^10^/L)	PT-INR	D-Dimer (mg/L)
Mild Illness	16	8:8	31.0 (24.5–64.5)	1 (6.3)	21.6 (16.9–26.7)	1.0 (1.0-1.1)	0.5 (0.3–0.7)
Moderate Illness	9	7:2	55.0 (26.5–70.0)	0 (0)	18.8 (13.1–20.8)	1.0 (1.0-1.1)	0.9 (0.4–2.9)
Severe Illness	16	5:11	52.0 (34.0–58.0)	1 (6.3)	21.0 (19.5–16.3)	1.1 (1.0-1.1)	1.0 (0.7–1.7) **
Critical Illness	5	1:4	60.0 (54.8–71.3)	0 (0)	15.8 (13.0–20.6)	1.2 (1.1-1.2)	3.4 (1.0–15.0) **

Data was shown as median (25–75 percentile). PT-INR, prothrombin time—international normalized ratio; **, *p* < 0.01 in comparison with Mild illness.

## Data Availability

The data presented in this study are available on request from the corresponding author. The data are not publicly available due to privacy restrictions.

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
