# Peer review of "Elevated Plasma Soluble C-Type Lectin-like Receptor 2 Is Associated with the Worsening of Coronavirus Disease 2019"

_jcm, 2022, doi:10.3390/jcm11040985_

Round 1
Reviewer 1 Report
Overall, the manuscript is poorly written. Some of the sentences are not correct or clear to me. For instance in the title: … associated with the risk of coronavirus disease. Does this mean: associated with an increased risk of COVID-19? or worsening of COVID-19 or….?
In the results section: the age was older in patients… etc. Maybe a native English speaker can be involved in the revision of the manuscript.
Abstract:
The background section is lacking. The methods section comprises only 1 sentences and needs to be elaborated.
Again, the abstract is written in poor English i.e mortality rate was 17% for other pneumonia should rather be: in patients suffering from other pneumonia and specify this. Lines 34-37 should also be rewritten (the words those should be replaced).
The Results section is lacking raw data (means/medians, significance levels, etc).
Introduction
There is a typo in line 53: COVID-19.
Material and Methods
A description of the study design is lacking.
Clearly describe the characteristics of the mild illness as this is not clear (various signs and symptoms?).
How was COVID-19 diagnosed?
Are the patients diagnosed with other infections admitted in the same period to the hospital?
How was the control group determined? Were they matched for age, gender, etc..
There is no information on the treatment the COVID-19 patients as well as the other groups received.
Results
In table 1, the total numbers of patients per group and percentages (gender) should be added. In the legend, abbreviations and description of the data (mean/median/range?) should be explained. Are *** and ** p-values in comparison with the unidentified clinical symptom? Please describe this, the same stands for ### and ##. The same stands for the legends of figure 1 and 2.
What was the cause of the deceased COVID-19 patients, if it was not attributed to COVID-19?
Discussion
The limitations of the study and suggestions for future studies are lacking, please add this to the discussion section.
Author Response
Comment 1. Overall, the manuscript is poorly written. Some of the sentences are not correct or clear to me. For instance in the title: … associated with the risk of coronavirus disease. Does this mean: associated with an increased risk of COVID-19? or worsening of COVID-19 or….? In the results section: the age was older in patients… etc. Maybe a native English speaker can be involved in the revision of the manuscript.
Response 1. This manuscript has been improved by professional editor who is a native speaker of English. The title was changed to “Elevated plasma soluble C-type lectin-like receptor 2 is associated with the worsening of coronavirus disease 2019. In the Results section, this sentence was changed to “The age of patients with other infections was older than that of those with COVID-19”.
Abstract:
Comment 2. The background section is lacking. The methods section comprises only 1 sentences and needs to be elaborated.
Response 2. The Background section has been now added. The Methods section has been improved.
Comment 3. Again, the abstract is written in poor English i.e mortality rate was 17% for other pneumonia should rather be: in patients suffering from other pneumonia and specify this. Lines 34-37 should also be rewritten (the words those should be replaced).
Response 3. These sentences including Lines 34-37 have been improved by a professional editor who is a native speaker of English.
Comment 4. The Results section is lacking raw data (means/medians, significance levels, etc).
Response 4. Raw data have now been added.
Introduction
Comment 5. There is a typo in line 53: COVID-19.
Response 5. Line 53: This has been corrected.
Material and Methods
Comment 6. A description of the study design is lacking.
Response 6. The study design has been now added.
Comment 7.
Clearly describe the characteristics of the mild illness as this is not clear (various signs and symptoms?).
Response 7. Various signs and symptoms have been now added.
Comment 8. How was COVID-19 diagnosed?
Response 8. The method used to diagnose COVID-19 infection has been added.
Comment 9. Are the patients diagnosed with other infections admitted in the same period to the hospital?
Response 9. This period has been added.
Comment 10. How was the control group determined? Were they matched for age, gender, etc.
Response 10. Control patients were managed in a period soon before the period in which the COVID-19 patients were managed. They were not matched for age, sex, etc. Because, the age of patients with COVID-19 infection is generally different from the age of patients suffering from other pneumonia. We have described this difference as a limitation in the Discussion.
Comment 11. There is no information on the treatment the COVID-19 patients as well as the other groups received.
Response 11. Information about the treatments has been added.
Results
Comment 12.
In table 1, the total numbers of patients per group and percentages (gender) should be added. In the legend, abbreviations and description of the data (mean/median/range?) should be explained. Are *** and ** p-values in comparison with the unidentified clinical symptom? Please describe this, the same stands for ### and ##.
Response 12. The tables including the legends, have been revised.
Comment 13. The same stands for the legends of figure 1 and 2.
Response 13. The legends for Figure 1 and 2 have been revised.
Comment 14. What was the cause of the deceased COVID-19 patients, if it was not attributed to COVID-19?
Response 14. The causes of death of deceased COVID-19 patients were added.
Discussion
Comment 15. The limitations of the study and suggestions for future studies are lacking, please add this to the discussion section.
Response 15. We added a “Limitations and further study” section.
Reviewer 2 Report
Dear Authors
I congratulate you for your work on sCLEC-2 and describing its significant roles in critical illness. The study definitely draws attention away from D-dimers which so far are being considered a tool to assess the severity among Covid-19 patients developing coagulopathy. I suggest here to kindly improve the methods section as it totally looks like your last contribution to mdpi (https://www.mdpi.com/2077-0383/10/13/2860/htm). I would suggest to reword the methods section and describe in detail the sCLEC-2 quantitative assessment. A visual conclusion of how sCLEC-2 is resposible/and where in the platelet activation that may be causing the severities independent of DIC grading might develop more ineterest among the readers.
Overall I support the study design and outcome.
Author Response
I congratulate you for your work on sCLEC-2 and describing its significant roles in critical illness. The study definitely draws attention away from D-dimers which so far are being considered a tool to assess the severity among Covid-19 patients developing coagulopathy.
Comment 16. I suggest here to kindly improve the methods section as it totally looks like your last contribution to mdpi (https://www.mdpi.com/2077-0383/10/13/2860/htm). I would suggest to reword the methods section and describe in detail the sCLEC-2 quantitative assessment.
Response 16. The Methods section has been revised in accordance with reviewer’s suggestion.
Comment 17. A visual conclusion of how sCLEC-2 is resposible/and where in the platelet activation that may be causing the severities independent of DIC grading might develop more ineterest among the readers.
Response 17. We added a schematic illustration demonstrating how sCLEC-2 is responsible for platelet activation independent of DIC grading.
Comment 18. Overall I support the study design and outcome.
Response 18. Thank you very much.
Round 2
Reviewer 1 Report
Dear authors,
Thank you for your extensive revision and responses to the raised comments. I still have one minor methodological error, i.e when reporting median (25th, 75th percentile): always do this in the same manner, for instance: 468 (355, 688) ng/ml and describe this in the statistical section.
Author Response
Comment 1.
Thank you for your extensive revision and responses to the raised comments. I still have one minor methodological error, i.e when reporting median (25th, 75th percentile): always do this in the same manner, for instance: 468 (355, 688) ng/ml and describe this in the statistical section.
Response 1. Data have been shown as “median (25th, 75th percentile)”.